# Promoting Water Efficiency and Hydrocitizenship in Young People's Learning about Drought Risk in a Temperate Maritime Country

**Verity Jones** [1]**, Sarah Whitehouse** [1]**, Lindsey McEwen** [2,*]**, Sara Williams** [2] **and Luci Gorell Barnes** [3]

1 Department of Education/Childhood, University of the West of England Bristol, Bristol BS16 1QY, UK; Verity6.Jones@uwe.ac.uk (V.J.); Sarah.Whitehouse@uwe.ac.uk (S.W.)
2 Centre for Water, Communities and Resilience, University of the West of England Bristol, Bristol BS16 1QY, UK; Sara3.Williams@uwe.ac.uk
3 Socially-Engaged Artist, Bristol BS16 1QY, UK; contactlucigb@gmail.com
* Correspondence: Lindsey.McEwen@uwe.ac.uk

**Abstract:** Engaging young citizens with drought risk and positive water behaviours is essential in domestic water demand management within the wider climate crisis. This paper evaluates a new research-informed, picture book—'DRY: The Diary of a Water Superhero'—that explores UK drought. The book's development was underpinned by research within the Drought Risk and You (DRY) project. The book's concept and storyline were co-produced by an interdisciplinary team, including a creative practitioner. This focused on key themes: drought definitions and types; drought myths; adaptation and young people's (YP) agency. Characters and storyline were co-created to promote YP's autonomy as change agents, and to encourage intergenerational and community learning. This paper evaluates the book from three perspectives: of YP, trainee teachers (TT) and teachers. Emergent themes are triangulated: drought as a sensitive issue, subject knowledge and changes in behaviour, and YP's misconceptions about drought and place. TT also contemplated their improved subject knowledge and barriers to engaging with positive water behaviours. Teachers reflected on classroom use of the book, prior experiences about teaching drought, curriculum context and st/age of YP engaged. This paper reflects on how these insights feed into school practice and water industry outreach, in developing effective learning resources that promote a valuing of water, behaviour change and wider hydrocitizenship among YP and their communities.

**Keywords:** drought; resilience; water efficiency; water behaviours; citizenship; agency; adaptation; young people; education; school; learning



## 1. Introduction

Over the past five years, the UK Research Councils have co-funded innovative, interdisciplinary research into the UK's past, present and projected future drought risk and its impacts. This aimed to step-change the evidence base to support better decision making in drought risk management in the UK, and to promote multi-sectoral engagement with drought as a hidden risk, including with different publics and communities. The Drought Risk and You (DRY) project (Dryproject.co.uk; accessed on 3 September 2021) worked to ensure that all sub-groups in communities have opportunities for knowledge exchange from the research, given the importance of engaging young people (hereafter YP) with environmental risks in a changing climate [1,2]. This involved collaboration between a socially-engaged artist, a geographer researching water risk, an environmental psychologist specialising in YP, and two expert teacher trainers to create a thought-provoking, research-informed narrative within a 'learning resource' targeting YP in primary school at Key Stage 2 (aged 7–11 years). Throughout this paper, we use the term 'young people' (YP), and position this age group as having agency and capacity to discuss complex

issues [3]. The picture book is entitled "DRY: the Diary of a Water Superhero", and was written from a YP's perspective. It aspired to bring awareness and adaptive actions for UK drought risk, and the need for positive water behaviours into the same frame to engage YP holistically about valuing water as 'precious, fragile, and dangerous' (see [4]). While the proposed audience was selected in line with the English national curriculum (Key Stage 2; 7–11 years old), the book has associated teacher notes that provide clear instruction regarding subject knowledge and vocabulary that can be adapted to suit the learner. In this paper, we focus on evaluating the content, impact and usability of the "DRY: the Diary of a Water Superhero" book.

There is growing concern about the role of the citizen in water management, and what hydrocitizenship might look like—in terms of personal reflection on self, relationship with others and environment, and care of place [5]. This involves looking at water through the lens of citizenship and vice versa. Such concern linking human agency and water as risk and resource is imperative within the climate crisis. This involves exploring the role of YP as eco- or hydrocitizens in learning for socio-ecological resilience [6], and in the development of individual and collective agency. Frequently, environmental education resources, designed to promote learning and agency among YP, may be developed without specialist knowledge of child development or learning, and/or without research underpinning, and may be used without evaluation [1]. This paper provides a multi-stakeholder evaluation (by YP; trainee teachers (TT); teachers) of the use and impact of the DRY picture book in English schools on drought risk awareness and water-efficient behaviours. This aspired to promote a deeper and more complete inclusion of YP's engagement with issues surrounding water security.

This paper aims:

1. To capture and synthesise the voices of YP of primary school age and teachers in their perceptions of drought and positive water behaviours through engagement with the DRY picture book.
2. To critically evaluate these reflections on using picture books as a resource stimulating learning for both resilience and hydrocitizenship, and the role of teachers in supporting this.
3. To reflect on what might be learnt from these children's and teachers' insights by water industry professionals involved in outreach using picture books to promote drought awareness, positive water behaviours and water efficiency through learning in primary school settings.

Given the importance of TT in cascading new knowledge in the classroom, and their potential to reach other and more experienced teachers, this paper appraises their perceptions of the impact, quality and usability of the book, as well as those of experienced teacher practitioners. Our targeted readership is those professionals working in water risk and domestic demand management who are tasked with or have an interest in engaging YP to promote learning for water efficiency and drought resilience in contexts of effective hydrocitizenship and changing climate. In previous research about flooding, we found that a wealth of resources (including narrative and digital games) have been produced (see [1]). We recognise this is multi-stakeholder territory—from those professionals with educational responsibilities in the water industry to NGOs working with YP in catchment settings. It can be difficult for those without training and experience to avoid a broad-brush, knowledge transfer approach in the engagement and dissemination of knowledge with YP (e.g., through whole school assemblies). This is due to a gap in appropriate learning resources, and expertise (and/or training) in how to use them. This paper seeks to address and bridge the gap between academic research and practice, sharing evaluative research with water professionals. This paper will initially provide a background context about UK drought at a time of climate emergency, and young people's cognitive and emotional development and ability to engage in issues relating to this. The wider DRY research project will then be considered, before we focus on the development of the research-informed picture book, DRY: the Diary of a Water Superhero. Following an outline of the qualitative methodologies used in our evaluation, we present data from YP, TT and teachers. We

then reflect on the development of research-informed resources for YP that can be used to support knowledge exchange between them and the adults that influence them in education for resilience.

## 2. Background Contexts

Here, we briefly consider why it is important to embed learning about UK drought as a hidden risk in formal (primary classroom) and informal learning situations. We recognise there are complexities in communicating with different audiences about the subject of drought (and its connotations with climate change), and the need for awareness and sensitivity about potential eco-anxiety. Crucially, we document the development of a research-informed (evidence-based) and evaluated learning resource for use with confidence by those aspirant to engage and promote YP's learning about the value of water.

### 2.1. Drought as a Risk in a Changing Climate

For many publics, Britain with its temperate climate is seen as green and wet, with infinite water supply [7], with flooding, rather than drought, perceived as the major risk. However, drought in the UK is a slow, diffuse, hidden and uncertain risk, involving multiple stakeholders that include different publics and communities. We have seen recent drought in 2011/12 in eastern England and summer 2018 (see UK Drought Portal (https://eip.ceh.ac.uk/droughts, accessed on 3 September 2021), impacting particularly on agriculture. Drought is also set to increase during the Anthropocene in the UK and globally [8].

There are many reasons why statutory agencies (water companies, environmental regulators) find it challenging to communicate with UK publics about drought [9], but it is important that they do, and that this includes YP. Drought and its diverse impacts are categorised in different ways, e.g., meteorological (rainfall) drought; agricultural (soil moisture) drought; hydrological (river) drought; water supply (or socio-economic) drought [10]. Each type of drought manifests itself differently and affects local communities in diverse ways, because some people are more water sensitised and impacted by drought than others [7]. In the UK, statutory authorities (water companies; environmental regulators) rarely use the word 'drought' because it is perceived as highly politicised. Public perceptions of drought as a rare and infrequent risk are reinforced by the news media, with weather reports tending to refer to positive elements of extended heat and lack of rain [7]. Drought memories and stories of extreme UK droughts that share local knowledge of impacts, along with coping and adaptive strategies, tend not to circulate in communities. Memories of the impacts of the 1975/76 drought, the most extreme in living memory in the UK, sit with a specific generation, now aged in their 60s. These memories are not necessarily negative; they can be of the halcyon days of youth, hot summers and sitting 'listening to Abba'. Intergenerational learning—learning from grandparent's stories—can be about lovely hot weather. If this is the only exposure YPs have to drought, the implications are potentially negative. However, deeper exploration of drought stories can reveal hardship due to past drought in the UK, e.g., in Cox's [11] book describing life as a Shropshire dairy farmer during the 1976 drought. Over time, memories become stories and may retain fragmented, interpretative information that can serve to provide intergenerational context and inform social learning. This context predicates the need for high-quality, accessible learning resources to promote awareness and adaptation to UK drought.

### 2.2. Climate Change and the Development of Eco-Anxiety in Young People

Climate change projections suggest an increase in the UK's future drought risk [12,13]. Stories about climate change and related environmental issues increasingly frequent media targeted at younger people (CBBC, First News), increasing their knowledge and awareness of these issues despite their lack of inclusion in National Curriculums. Drought may not be a familiar experience for YP; they may experience early stages without even knowing it is happening, and media coverage aimed at YP (and adults) can be scant without a visual 'hook'. Those involved in helping YP gain the knowledge, skills and dispositions to

respond resiliently to these risks need research-informed learning resources, confidence and training in order to do this effectively. Child psychiatrists report that a half of all clients they see are suffering eco-anxiety [14], with the Royal College of Psychiatrists [15] noting that YP find their anxieties about the state of the environment difficult to manage due to the enormity of the problems and feelings of helplessness. Recent surveys [16,17] have highlighted the concern YP have for the environment in the UK, as well as a desire by YP to learn more about the climate crisis.

For those who have engaged in the theorising and practice of teaching about climate change, and the related environmental consequences (drought and water shortage being just one example), have for many years considered the pedagogy of hope to be important. Freire [18] regarded hope as something we cannot exist without, noting that 'without a minimum of hope, we cannot so much as start the struggle . . . Hence the need for a kind of education in hope' (p3). Here, the educators' role is primarily to evoke and guide students' hope. In contrast, Snyder [19] argues that hope is not inherent, but a learnt cognitive process. In both accounts, the educator's role is essential to development of a sense of hope. YPs need the facilitation of explicit learning opportunities on how to reason in this way and support for this with more goal directed thinking for behaviour change [20]. Taking action in response to the negative consequences of climate change can help individuals feel a sense of control within the complexity of this 'wicked problem' [21], given that every drought is unique with cascading impacts, and the need for multi-stakeholder engagement in finding solutions. Identifying, planning and taking adaptive actions can build hope and resilience [14].

However, Hicks [22] and Ojala [23] warn that simply 'being hopeful' may not necessarily see behaviour changes that will counter the negative impacts of climate change. They note two possible outcomes: the development of hope that the reality is wrong (and so the individual denies the situation they find themselves in and carries on with their current behaviours), and the development of hope that they can change the future (and so alters their behaviour). With this second outcome in mind, we consider how a picture book, developed specifically with the intention to support YP in understanding the complex connections between drought and positive water practices through increased hydrocitizenship, might facilitate knowledge acquisition, reflection and behaviour change.

*2.3. Sustainability Agenda in Education and 'Learning for Resilience'*

The UK Government, along with a range of national agencies, signed up to a strategy in March 2005 that aimed to integrate the principles, values and practices of sustainable development into education [24]. While there is a global policy focus on embedding education for sustainable development [25], national policy sees a more patchwork approach very much driven by individual government priorities. For example, following a Supreme Court ruling in 2003, Indian government agencies produced extensive content on environmental education. This resulted in over 300 million students in 1.3 million schools receiving some environmental education training [26]. In Costa Rica, environmental education is fully integrated and prioritised within school curricula [27], while Italy may be the first country to require climate change education in all schools [28]. Here, in Great Britain, both Wales and Scotland have embedded the UN Sustainable Development Goals (e.g., SDG 13 Climate action) into their national curriculums [29,30], and see this as an entitlement for learners, and a core part of teachers' professional standards.

Of course, it is not only within formal education that climate education is practised. Scotland has seen a range of initiatives established to encourage local communities to engage with sustainable development through its Climate Challenge Fund. Since being established in 2008, £111 million has funded over 1150 projects in 32 local authorities [31]. As part of its Climate Action Fund, the National Lottery has funded 23 projects across the UK at a cost of £19.5 million. While this explicit engagement and citizen facilitation of action around climate change goes on, England's most recent primary National Curriculum [32] saw the omission of explicit reference to sustainable development and environmental

education (and many other related terms such as climate change). Water is a theme in the Key Stage 2 geography and science curriculum (7 to 11 year olds), but teachers embed this theme in ways they feel appropriate, and in line with their own subject knowledge surrounding the issues. There is no guidance as to appropriate pedagogy and the territory of hydrocitizenship is not a mandatory theme.

### 2.4. St(Age) Development (Cognitive Development) of YP's Learning

The st(age) at when to introduce YP to topics such as drought caused by environmental changes as a result of climate change is not well understood and can be contentious [33]. In such instances, some believe that YP should not learn about or be involved in these issues as they are not only invisible in the curriculum, but they have concerns that YP should be protected from these complex and worrying issues [3]. In this context, childhood is considered a social construction and YP are positioned in need of protection from complex issues, as opposed to positioning YP as capable, ambitious learners who have agency in their behaviour. This could also influence the development of YP's knowledge, skills and dispositions as they mature. In addition, there are considerations regarding when YP are cognitively and emotionally able to engage in these difficult concepts, and the role that affect and empathy may play in knowledge acquisition, understanding and potentially action.

Piaget [34] argues that from 7 to 11 years, there is a shift in dominance to logical thought, with YP beginning to use inductive logic and reasoning—both skills essential for critical thinking. In addition, Bruner [35] and Vygotsky [36] acknowledge that while YP may as individuals find such learning challenging, with the aid of a more knowledgeable other (teacher or peer or scaffolding through resources), engagement with, and development of, understanding is possible. In this case, the storybook can be seen as such a resource—created by a group of interdisciplinary experts and embedded within extensive evidence-based research. It is widely documented that there is educational value in picture books, such as cultivating kindness, promoting identity and stimulating cognition [37–39]. However, to date, there is a gap regarding the relationship between picture books and promotion of hydrocitizenship. The link between reading picture books and the acquisition of environmental knowledge has been identified [40]. However, while there have been books for young readers about resilience to extreme weather (flooding, heat, snow, e.g., 'Susie the Childminder' books; Hampshire Fire and Rescue, UK (https://www.hantsfire.gov.uk/kidzone-and-schools/susie-the-childminder/ accessed on 3 September 2021), there have, to our knowledge, been no evaluations or impact assessments of these. Increasing drive exists to engage YP about water efficiency in schools (e.g., Waterwise (www.waterwise.org.uk accessed on 3 September 2021) campaigns and water company activities). Hsiao and Shih [41] note that using scientifically informed picture books could help YP understand their everyday environment. Here, we evaluate that experience in relation to knowledge acquisition, hydrocitizenship and adaptive resilience for both young learner and facilitator.

Growing interest exists in what learning for resilience might look like in different contexts including with YP [2,42]; what knowledge, skills and dispositions are important. What is clear is that knowledge is not enough, and that intrinsic personal factors such as value systems, emotion and affect and sense of place are critical in bridging the knowledge–value–action gap. This poses questions about what resilience as practice [43] might look like in water education for YP, and reinforces the value of considering the water environment holistically and systematically with different publics (i.e., risk and resource together).

### 3. The Drought Risk and You (DRY) Project

The four-year Drought Risk and You (DRY) project (funded through the UK Natural Environment Research Council) aimed to bring science and stories together to support better decision making in drought risk management in the UK [44]. The research methods involved both specialist scientific modelling and gathering science-stimulated stories of

drought within seven catchments across hydrometeorological and urban-rural gradients across the UK. This involved collecting stories across sectors including environment, health and wellbeing and public/communities [45]. DRY recognised the importance of approaching the topic obliquely to garner stories of drought [46], and to link 'drought thinking' to discussion of wider water relationships and adaptive strategies. Memory of severe drought in the UK, and related local knowledge, tends to be partial and unconnected to future preparedness and action. The evidence base garnered by the DRY project highlighted a series of drought myths that are pervasive in the British psyche, and important in terms of grey areas in citizens' understandings of UK drought risk. Within DRY's case-study catchments, drought is an environmental hazard projected to increase (e.g., [47]).

The development of the DRY picture book was underpinned by original interdisciplinary research within the DRY project. (The book was launched at the 'About Drought Download' event at the Royal Society, London in November 2019. See https://aboutdrought.info/about-drought-download-nov-7th/ accessed on 3 September 2021). It drew on research linking different kinds of 'evidence'—science and stories from seven river catchments across the UK—alongside cross-disciplinary expertise (e.g., [46,48]. We focused on four themes informed by the research and worked to weave them through the book's narrative. First was the classification of UK droughts and identification of their impacts. This involved exploring 'what is drought', recognising that drought means different things to different people depending on its impacts on humans and non-humans—both wild and companion animals. The book identifies Wilhite and Glantz's [10] different types of drought, and how they might manifest themselves locally. Second, the book aimed to interrogate common misconceptions surrounding UK drought: it takes pervasive myths about UK drought and water availability and challenges them (e.g., that water is infinite and free, and that drought only happens in summer). Third, the book works to investigate what actions UK citizens (within families and communities) might take to prepare for drought. It asks about the possible adaptations and changes in water behaviours, including saving and storing water, thinking about water stored in the production and processing of food (hidden or embedded water), and the differences between directed actions to save water (turning off a tap) and changes in social norms (normalising use of harvested water). Fourth, it aims to explore the civil agency of YP and how they might act as change agents in rethinking water behaviours in their families and communities. The narrative involves peer-to-peer and intergenerational learning.

The picture book, its concept and storyline were co-produced over nine months by the interdisciplinary authoring team (see earlier). In addition, detailed teachers' notes were developed comprising of subject knowledge (including a glossary of terms) and a range of cross curricula activities that supported the picture book. The idea is that the book could be used to teach a subject directly related to the curriculum (for example, mathematics) whilst the subject matter (and learning) is about drought. A further key focus of the teachers' notes was an action plan that enabled young people (with support from their teacher) to put change into action, as this is a key pedagogy required to support eco-anxiety [49]. The layered illustrations in the picture book, integrating found objects, were co-developed as an integral part of that co-authoring process. This expertise combined to provide a unique skill set for the book's development. The co-authoring process involved five creative authoring meetings that were audio-recorded, along with virtual exchanges of ideas, resources and drafts. Preliminary work involved a content analysis of the UK primary curriculum, and a review of picture books for YP that integrate fact and fiction. The storyline of the book, written as a young girl's diary, runs monthly over the course of a year in a city such as Bristol, UK. It tells how an ordinary schoolgirl (approx. 10 years old) in the UK transforms into a water superhero when a dry summer and winter with little rainfall lead to drought. Seeing life through 'water goggles', the girl shares her new-found understanding and respect for water with her family. On her stimulus, family members start to change their lifestyle habits. She then works to share this love and understanding of water with her school and community, as the drought progresses. Throughout, we chose actions that YP

could do themselves in order to emphasise the message of personal agency (for example, pinpointed actions such as having shorter showers, turning the tap off when cleaning teeth while also questioning social norms of water use). Through the narrative, visual clues and activities, the resource explores how YP can be key hydrocitizens and influencers at home and in the community.

The book is designed with YP's cognitive st(age) in mind (7–11 year olds), and positions them as agents of change, with 'pester power'. Consideration is given to the overall messaging in terms of empowering YP to act, intergenerational learning and influence, vocabulary, and mathematical abilities—all contained within the overall contextual research findings about drought (e.g., drought myths) from the wider DRY project. Key was relevance to the national curriculum to encourage usability. In terms of physical character, we paid attention to the size of book and print quality, recognising this has a large impact on appeal to young readers, and to adults using the book. The book gained a publishers' silver award from the UK Geographical Association (http://dryproject.co.uk/narratives/book-bringing-drought-research-to-children-wins-national-award/ accessed on 3 September 2021) and has since been included in a Parliamentary research briefing report [50].

## 4. Methodology

This research provides evaluative data and a platform for YP's voices that are often silent due to an absence of evaluation of content, impact and usability of resources. While we have not used a formalised evaluative framework, we have drawn on the principles of democratic evaluation [51]. This approach enabled us to 'give voice' to those YP and investigate what they thought about this new resource (the book), what they noticed and learnt about drought. Often YP are not asked these questions about the resources that are used to engage them in learning. A mixed method approach was used to gather perceptions about the book's use and its impact across three key target groups: YP, teachers and TT. Garnering the perceptions of TT was considered critical to the inculcation of new themes for primary school teaching in the context of changing society, extreme weather risk and the climate crisis. TT are the next generation of educators, and those who have had direct experience of resources and teaching pedagogies are more confident and effective in successfully delivering opportunities for high-quality environmental learning with their future pupils [52]. On training placement and in their first teaching posts, they are able to cascade new learning resources and practices into the classroom and schools, themselves acting as agents of change.

Data collection took place in the latter stages and after book development. During development, draft books were tested using participatory methods with YP of primary school age and teachers in two Bristol schools, and were also informally tested with parents and YP. During the formative evaluation of the book, participatory workshops took place with 9–10 year olds (n = 60) and interviews with their teachers (n = 3) about their reflections on how the book might be used and improved. Feedback from pupils and teachers informed revisions to the final copy. After publication of the book, multi-methods of summative evaluative were used: two focus groups with 9–10 year olds in a class setting (n = 12); included mapping YP's emotional responses to the book drawing (on an agreed bank of words); a questionnaire survey of TT (n = 90); and three in-depth, individual, semi-narrative interviews (ca. 30 min) with teachers who had used the book in practice.

Focus groups and participatory workshops were used as methods of data collection with the YP in order to facilitate an opportunity to listen to and understand how participants' feelings, opinions and thoughts were formulated [53]. During the focus group, YP mapped their emotional responses to the book. The vocabulary used by YP was discussed and agreed on amongst participants. This was to ensure that all participants were confident with meaning and felt they were able to represent their own response as closely as possible. Emotion was mapped at the points participants felt most appropriate as a group.

Ninety TT on a one year, postgraduate course, completed the online questionnaire survey in the first term of their studies. This cohort was drawn from a range of career

changers and academic disciplines (including educational settings as teaching assistants, 1:1 child support and other roles within the school system). This captured responses on the following themes: whether drought was a topic taught or observed in the classroom, how this might be approached and how the book might be used, as well as any impacts on participants' perceptions of drought and water use gained as a result of engaging with the text and imagery. This data gathering was followed up by a focus group.

The data were analysed using a thematic approach, following Braun and Clarke's five-step system [54]. This began with researchers becoming familiar with the data; generating initial codes; searching for themes; reviewing the themes; refining the themes; and looking for commonalities and complexities, which are presented in Figures 1–3. Triangulation of themes between the three groups was then undertaken. Ethics permission was obtained from the University of the West of England, Bristol in line with British Education Research Association's guidelines [55]. Consent was gained from all teaching staff, with schools acting as brokers for YP in the data collection.

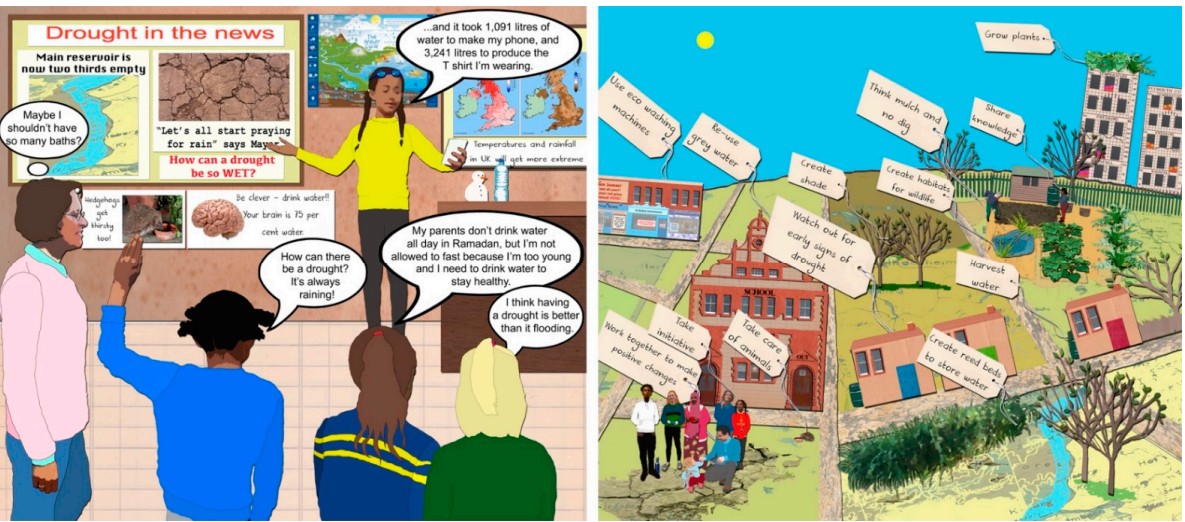

**Figure 1.** Two images from the DRY picture book (Luci Gorell Barnes, 2019, digital collage). The link to the full online book is given in the 'Data availability Statement' section.

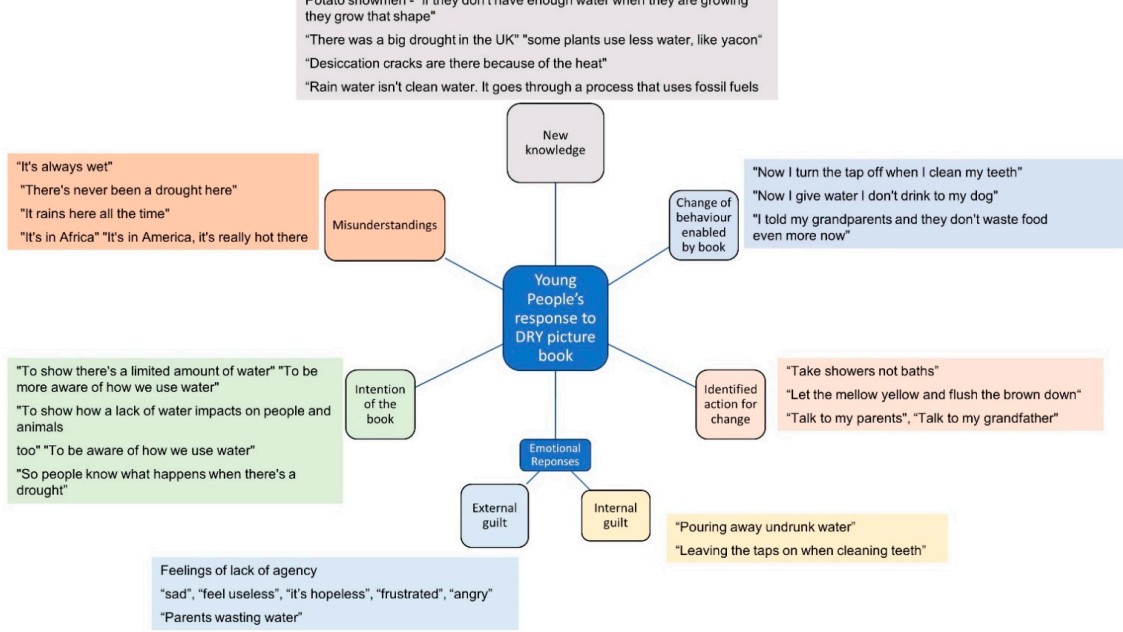

**Figure 2.** Young people responses to the DRY picture book.

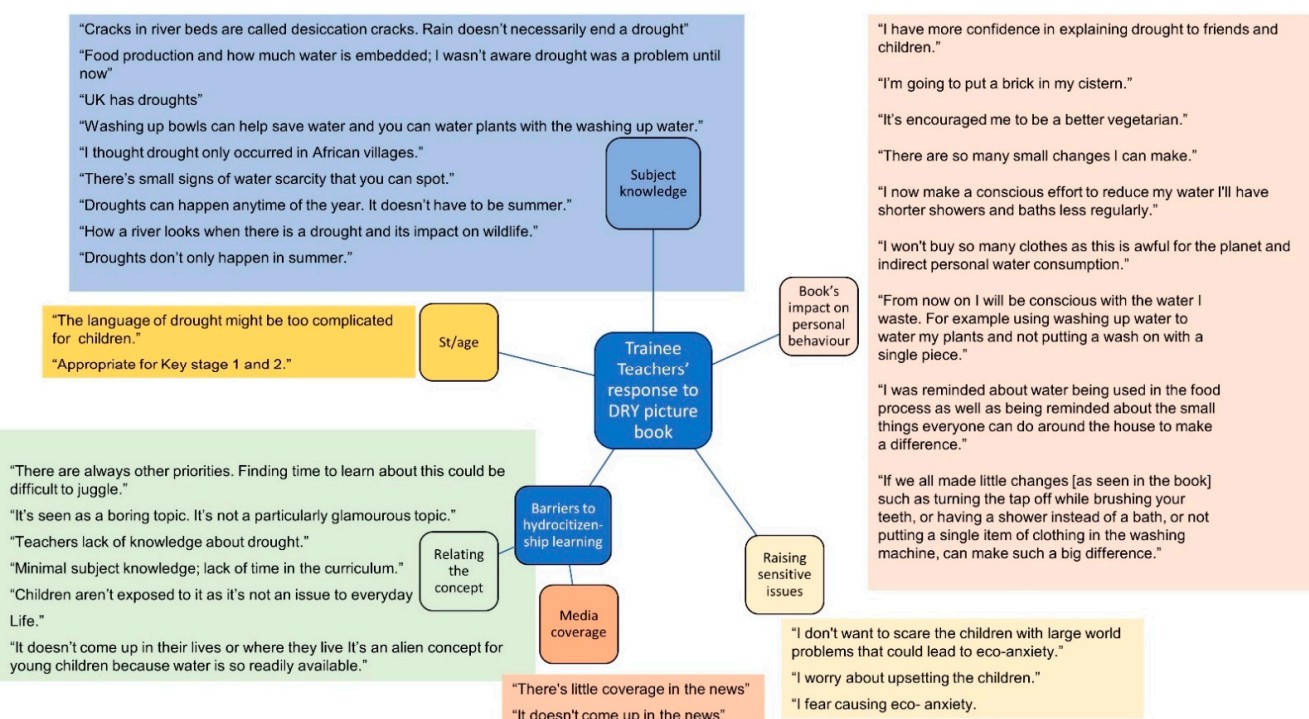

**Figure 3.** Trainee teacher responses to the DRY picture book.

## 5. Results

The results of the thematic analysis are presented in sections focusing on YP's voices alongside those of TT and teacher practitioners. A discussion bringing together these themes and highlighting commonalities follows.

### 5.1. Young People's Voices

Three themes were identified through data analysis: concerns surrounding drought as a sensitive issue and it causing anxiety; the development of subject knowledge of resource; and YP's agency for behaviour change. Figure 2 shows a thematic mapping of YP's responses to the DRY picture book.

5.1.1. Drought Risk as a Sensitive Issue

The emotional responses YP felt as they read the book followed the narrative pattern (Table 1). In September, the protagonist was surrounded by a dry climate and readers begin to recognise the problem of water scarcity. In November, she begins to find out more about water usage and wastage within the home and garden, which raises YP's concerns—some recognising that they waste water in similar ways and so beginning to feel guilty as they reflect on their own water behaviours. In December, the protagonist learns more about embedded water in food and everyday goods through her Christmas celebrations and feasting. This knowledge once again is met with negative emotion from the young readers; some negativity is directed at how others behave and worrying about this, others reflecting on their own similar behaviour around food waste and feeling guilty. By February, when the protagonist is making progress with her class project about water and feeling more positive, the readers also find more hope. By March, the story shows that the initial burst of enthusiasm is lost, and our main character cannot change people's water behaviour on her own. In reaction to this, the readers also feel a sense of frustration. The final section of the book leads the young girl to working towards making small changes in the home and

garden, as well as working with her local community for wider adaptation surrounding water usage. A majority of the readers by the end of the book reported that they felt they too wanted to do something, and could make a difference by improving their current water behaviour.

**Table 1.** Examples of the emotional responses to the narrative in the DRY picture book.

| | Month in the DRY Picture Book (Runs Over a Year—September to August) | | | | | |
|---|---|---|---|---|---|---|
| Participant | Sept | Nov | Dec | Feb | March | The End (Aug) |
| 1 | Guilty, want to do something to change it, sad | Useless, frustrated | Angry, guilty | Relieved that something is being done | Angry that no one is helping | Positive, want to do something to change it |
| 9 | n/a | Frustrated | Annoyed Guilty | Positive, happy, want to do something to change it | Bored, frustrated angry | Positive, happy, hopeful |
| 10 | n/a | Worried, guilty | Worried, guilty | Frustrated, guilty, sad, worried | Frustrated, worried | Want to do something to change it, hopeful, happy |
| 12 | Sad | Worried | Guilty | Positive | Frustrated | Want to do something to change it |
| 13 | Empty | Worried | Hopeless | Happy | Useless | Want to do something to change it |
| 14 | Sad | Hopeful | Jealous | Happy | Bored | Delighted |

This use of emotive language indicates that the discussion of drought with YP can incite feelings of a sensitive nature, especially around guilt. This may impact on a teacher's willingness to enter into conversation, leading to an avoidance of the issue, meaning that learning resources such as this are required. The book provided an opportunity to observe various water practices and this facilitated an opportunity to reflect on their own behaviours. For example:

> *"I sometimes feel guilty because I might pour a glass of water and then pour half of it away."* (YP2, focus group A)

> *"I leave the tap running. I feel a bit guilty about that now."* (YP5, focus group A)

As well as the feelings of personal guilt, the book incited one participant to reflect on the need for collective action:

> *I feel guilty just because I'm one of those people who could be trying to help but many people are actually doing that, but I'm one of those people who is doing it . . . trying to help. There's just not many of us. It's a bit like a boomerang. You throw the boomerang and it just comes back to you. You need to pass it on around the world.* (YP 9, focus group B)

Here, the book provided opportunities to discuss the need for collective action for change. The YP felt a sense of guilt on behalf of a larger group; guilt that they were not able to make an impact, and awareness of global scale.

5.1.2. Subject Knowledge and Behaviour Change

Young people said that the book was engaging, '*filled with lots of ideas*' about water usage and how to save water. For example:

*"I now know there are plants that don't need as much water, like yacon."* (YP 1, focus group B)

*"I'd never really thought that the UK could have a drought . . . I've learned a lot"* (YP 6, focus group A)

In this case, a link to the impact of drought resistant plants was made. However, in some cases, connection to wider impacts was identified, as articulated below:

*"Rainwater isn't clean water, it has to go through a whole process that uses fossil fuels, that if we're wasting the water, we're wasting the fuels and adding to pollution and global climate change. It's really bad."* (YP 3, focus group B)

Some YP were then able to take the new knowledge, recognise the potential impact and identify personal responses regarding their own water practices. For example:

*"Sometimes I leave the tap running when I'm brushing my teeth. Now I would turn the tap off [having read the book] rather than let it run."* (YP 1, focus group A)

*"Because the book told me that like, turkey uses like, 40 litres of water, 400,000 litres go into one single turkey. We can't just throw it away."* (YP 6, focus group A)

This second example highlights how the YP has not necessarily quoted the correct statistics but is very much aware of the scale of the problem. In addition to this, there was also discussion that related to YPs agency as communicators of/for change:

*"My grandfather had 2 sons, my dad and my uncle, and they all hate wasting food, they hate it. Hate it. And I told them all about how much water is wasted too, and now they hate wasting food even more!"* (YP 6, focus group B)

This example highlights that not only did the book share knowledge of water practice with the reader, but promoted awareness of intergenerational knowledge.

5.1.3. Young People's Misconceptions about Drought and Place

During the focus group B discussion, participants considered where the book was set:

*YP 9: "I think it's in Africa. In Africa it doesn't rain that much and there's cracks in the floor.*

*Interviewer: this was actually set in the UK.*

*All: What? Really?*

*YP 6: I don't think I believe that. It rains a lot here.*

*YP 1: and it says in the story there's been a drought and there's never been a drought in England."*

Some YP's comments indicate that they have misconceptions regarding drought and water scarcity in the UK, which the book was unable to rectify. There is an assumption that the UK is too wet and drought only happens in other, warmer, distant places ('othering' drought). This mirrors research findings with adults in the DRY project. However, despite this, YP noted that they were still able to relate to the story because "*this story is more about normal life*". They spoke about how having a young protagonist in a recognisable urban setting allowed them to engage with the narrative and the messages surrounding water behaviours. Recognising such behaviours in the book that were familiar allowed readers to reflect on their own practice without having to recognise similarities of place. Others observed their learning:

*"I'd never have thought the UK could have a drought . . . it's weird, I thought it wasn't possible. I've learned a lot."* (YP 9, focus group B)

This suggests a positive role for early interventions to increase awareness of UK drought risk, but with on-going reinforcement activities. It also confirms that same or similar myths to adults pervade YP of this age. It could be that these myths are developed

early in family settings, and then are retained. This gives enhanced reason to engage YP in the topic and make available effective research-led, evidence-based, learning resources.

*5.2. Trainee Voice*

Seven TT felt confident teaching about drought in the UK, while 95% had not seen or taught any lessons about drought or water scarcity. This may not be surprising considering trainees were in the first few months of training. Similar themes were identified in the data analysis for both trainees and YP (Figure 3). Whereas the YP communicated their emotive feelings about drought, the trainees reflected on their concerns surrounding teaching about drought and initiating an emotive response within the YP.

Trainee teachers considered the teaching of water practices and drought to be a sensitive issue and a possible cause for anxiety in the classroom. This suggested they had an awareness of increasing levels of eco-anxiety. Similar to YP, TT also discussed misconceptions of the issues around drought, and a lack of subject knowledge. However, further themes identified in this data included TTs' reflections about media influence on understandings, lack of curriculum inclusion, alignment and the time to teach these issues, and teacher agency for behaviour change.

5.2.1. Drought as a Sensitive Issue

Trainee teachers raised concerns regarding the emotional impact teaching about drought might have on YP in their classrooms, and considered this as a barrier to them engaging in the topic. Many commented similarly to the following:

*"I fear causing eco anxiety for the young people—it can be distressing for them to learn about [drought]."* (TT 24)

*"I don't want to scare them."* (TT 88)

This highlights trainees' assumptions that YP are in need of protection with regard to the impacts of drought and water scarcity. This anticipation of a difficult emotional response that as an educator they have to negotiate, could be a barrier to engaging with this issue. This is in contrast to YP's actual responses to the issues raised in the book; while eliciting an emotional response, the journey through the book left readers feeling hopeful and empowered to change.

5.2.2. Trainee Teacher Improved Subject Knowledge

Having read the book, TT were able to identify a wide range of new knowledge about water and water practices they had acquired from the narrative. Examples include

*"I was reminded about water being used in the food process as well as being reminded about the small things everyone can do around the house to make a difference."* (TT 7)

*"I was very naive to the situation and only thought drought happened in African villages."* (TT22)

Similar to YP, some TT were able to embed their new knowledge into changes in water practice that they could do themselves.

*"Washing up bowls can help save water and you can water plants with the washing up water."* (TT 38)

*"If we all made little changes [as seen in the book] such as turning the tap off while brushing your teeth, or having a shower instead of a bath, or not putting a single item of clothing in the washing machine, [this] can make such a big difference."* (TT 36)

This indicates that both adults and YP, without prior understanding of the idea of hydrocitizenship, are influenced through the reading of the book with the potential to move from knowledge to action and enable the identification of positive behaviours.

### 5.2.3. Barriers to Engaging with Practices of Positive Water Behaviour

Many TT acknowledged that they had little subject knowledge regarding UK drought and positive water practices before reading the book. However, personal knowledge about the importance of hydrocitizenship was not the only barrier recognised to learning and action. Many trainees saw the media as a barrier to engaging in this topic due to the perception of its neglect. Trainees talked about the media in terms of it having "little coverage", and UK drought "not being covered" in the news, but with a focus on drought in faraway places (drought 'othering').

This lack of coverage was perceived by many as a reason to label the topic of drought and water scarcity in the UK as one that may be uninteresting to YP. Trainees repeatedly commented that drought was 'boring', for example:

*"It's a boring topic, it's not particularly glamorous."* (TT 61)

Trainees also commented that the demands on curriculum time is also a barrier to including drought and water scarcity into the classroom:

*"There are always other priorities."* (TT 14)

*"Finding time to learn about this could be difficult to juggle."* (TT 46)

This indicates that framing and presenting the topic of water scarcity and drought is important. Using a storybook, with vibrant illustrations visualising 'hidden drought', provides an opportunity to overcome these barriers with non-experts, and to align the content to different curriculum subjects. (The accompanying teacher's notes - not evaluated here - are aligned to curriculum subjects. These can enable the subject of drought to be taught through an English lesson, for example.)

### 5.3. Teachers' Reflection on Using the Book in the Classroom

The three interviews with teachers, who had used the book within their classroom practice, raised the themes of subject knowledge, previous experience of teaching drought/water scarcity in the National Curriculum [32] and whether the book was st/age appropriate (see Figure 4).

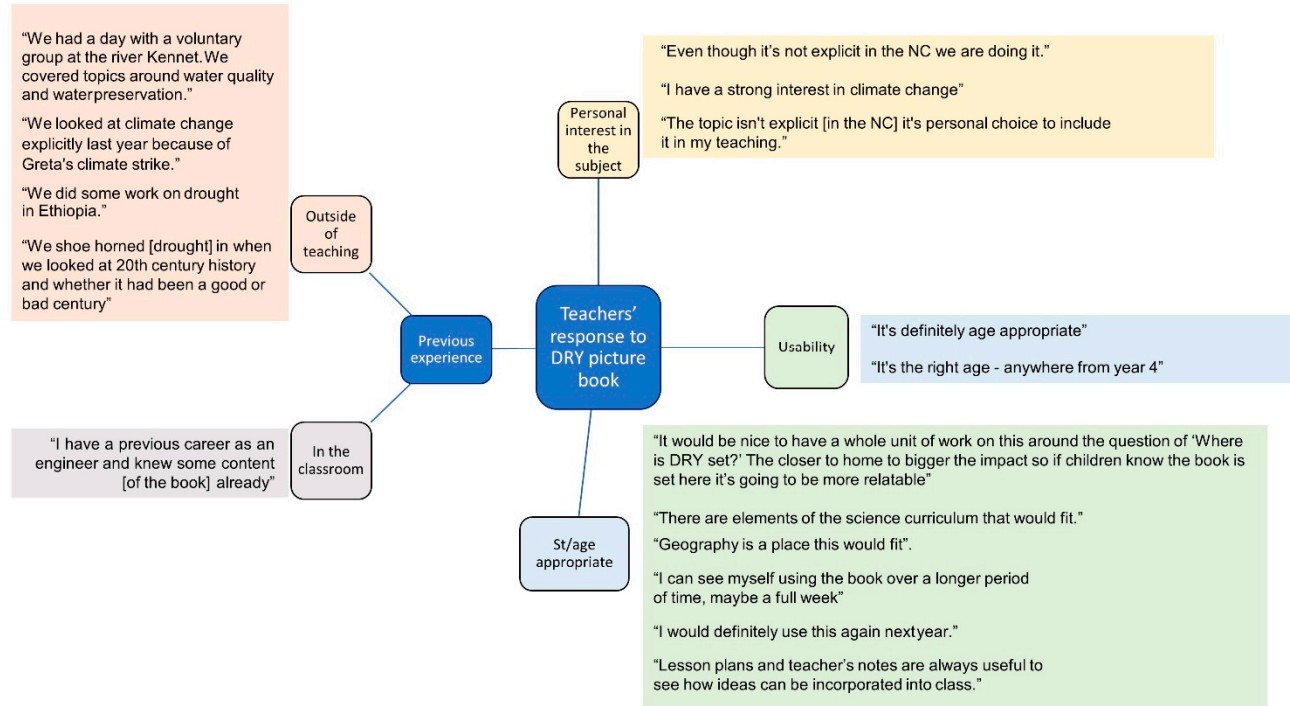

**Figure 4.** Practicing teacher responses to the DRY picture book.

Teachers commented:

*"This book and associated activities has really made me think about how I use water and I know it's a resource I'll be using again and again."* (teacher 2)

*"A great story that supports geography and science topics."* (teacher 3)

5.3.1. Previous Experience of Teaching Drought

Two teachers had experience of teaching about drought within Key Stage 2. One had taught it in the UK context (with reference to local river studies), the other with reference to droughts in distant places (in this case, Ethiopia). It was interesting to note that one teacher commented on how current, local activities had influenced the incorporation of climate change education in their classroom the previous year.

*"We looked at climate change explicitly last year, I think because of the climate strikes."* (teacher 1)

In this example, the teacher responded to the immediate learners' interests as Greta Thunberg had visited Bristol and some YP within the class had been on the climate strikes. One teacher, who had not taught about drought before, was keen to engage with the DRY resources as she commented:

*"The children are so interested in the latest climate marches I feel I really need to engage with this issue and this book will be really useful to do that."* (teacher 3)

This indicates that it is important for educators to recognise key times when learning for hydrocitizenship may be capitalised upon.

5.3.2. Curriculum

All teachers commented that even though there was not explicit reference to climate change, drought or water scarcity within the National Curriculum [32], they all taught it to different degrees, mentioning how they "don't teach it explicitly" and have to "shoehorn" it in.

All teachers recognised that the inclusion of climate change, and how they incorporated this in children's learning, was very much down to personal teacher's preference as there is no directive as to how to teach this.

*"It's personal choice to include in my teaching."* (teacher 2)

*"Even though it is not explicit its personal choice to include in my teaching."* (teacher 3)

It seems that in current times, the personal interests of the educator are essential to mobilise, if principles and practices of hydrocitizenship are to be embedded within YP's learning and behaviour change enacted. The question, under this model of curriculum, is what are the consequences for YP who do not come across a teacher with these interests during their education?

In addition, one teacher commented that:

*"It is the personal interests of the children that is urging me to take a look at this theme."* (teacher 1)

This response reveals a degree of YP's agency—the YP are able to influence the teacher to find out more about a topic. This may or may/not be widespread across schools, and may be contingent on school culture and/or individual teacher's initiative and practice. It may also depend on the YP getting knowledge/interests from outside school (e.g., media; agencies such as water professionals involved in schools outreach).

Once again, this indicates that meeting the needs of learners by providing opportunities for time appropriate learning is essential. For schools (and water professionals), having resources that link with the National Curriculum [32] is useful, in that it may provide a purpose for teaching and engagement.

### 5.3.3. St/age

Teachers commented that they felt the book was age appropriate and recognised that it could be used with both younger and older learners (8 to 13 year olds).

*"I think it's age appropriate, though you could use it lower or higher."* (teacher 1)

*"I think it was the right age, anywhere from year 4 to year 8. [ages 8–11]"* (teacher 2)

No teacher considered the theme of climate change or drought/water scarcity as being inappropriate for their pupils, although they recognised sensitivities surrounding the topic and possible links to climate anxiety.

*"Even though drought might be seen as a scary subject none of the children were scared by the story. They really liked it and could relate to what was going on."* (teacher 1)

This indicates that the narrative of the story is recognised as one not facilitating fear, but as retaining a hopeful tone in creating relatable role models for hydrocitizenship.

## 6. Discussion

### 6.1. Developing Learning Resources for Knowledge Exchange

This learning resource forms a unique part of knowledge exchange about UK drought, positioning YP as young citizens with agency. With climate crises and increased eco-anxiety amongst YP, this resource provides teachers (practising and trainee) and water professionals who tasked with engaging YP, with a tool to support understanding the complex issues surrounding the often hidden risk of drought in the UK. The book's originality comes from content and approach. The challenge was to make academic research accessible to YP, and to unpick visually the four themes identified and explored in the resource in engaging ways. In doing this, we deliberately interwove the narrative and developed the visuals with thought-provoking facts. In promoting usability, layered illustrations were developed to act as stimuli for discussion, and to promote thinking around individual and collective agency. Chawla and Cushing [56] consider how action for the environment often looks at private behaviour change (for YP). They argue that solutions need to be multi-faceted, and combined with actions for collective change amongst the public. The book looks at, and connects, both the personal and community sphere.

Our multi-stakeholder evaluation indicates that there is significant value in combining consideration of water as a risk (drought messaging) and promoting positive water behaviours (water as a finite resource) in the same learning intervention. This brings the sensitivities of risk and the positive ethos of agency into the same frame. Drought as a hidden risk is not easy territory for engagement. However, feedback from all groups was that the resource visualised risk, engaged and encouraged questioning. Despite the lack of explicit focus in the National Curriculum [32], we have seen that there is an appetite from both trainee and practising teachers to engage with this challenging issue. This is being led to a certain extent by YP who are influenced and knowledgeable (through media/school climate strikes/Greta Thunberg/David Attenborough), providing further evidence of YP's agency and potential to influence. There is also strong potential for teachers and practitioners to learn alongside YP about UK drought, and for that learning to be mutually reinforcing across both groups.

It is always challenging to evaluate the impact of a one-off intervention. We have undertaken no follow up of the persistence of the YP's awareness that drought is a UK risk or evaluation of YP's 'intention to adapt' their water behaviours. There is potential to return to a longer-term evaluation in the future. However, we have observed that the book promotes the idea of personal responsibility to make small changes that could be followed up in school, home and in cross-curricular activities. In terms of the impact of an intervention, we argue the value of training new teachers in using the book in their teaching practice as a sustainable way forward. It maybe that water professionals will be able to have further conversations with teachers about 'watery topics'. Teachers may also be more knowledgeable and receptive to working with water professionals, and to developing this focus in class.

Working with TT in the book's development has meant that we have been able to explore the value of those connections longitudinally. Such a model could be promoted in other teacher training courses in HEIs nationally. While citizenship education forms part of the school curriculum, there is opportunity to integrate eco- or hydrocitizenship in the learning and development of YP. Learning about drought brings challenges, but also creative opportunities. Building on the experiences of using the DRY picture book with TT, those involved in informal learning (including water professionals) could also utilise the book in their engagement strategies with YP. HEIs may be able to offer CPD/short-training sessions for water professionals in the use of such learning resources to develop expertise in knowledge and delivery and encourage meaningful engagement.

*6.2. The Value of 'Picture Books' in Learning for Resilience*

This research has shown that both YP and those who facilitate their learning are able to use the DRY picture book to engage and enable knowledge acquisition. The book is a resource that can advocate a sustainable approach to water resource efficiency and resilience-building to risk, and begin thinking about positive actions in hydrocitizenship (sense of self, concern for others and environment; [5]).

TT responses indicate that the book is able to provide a non-threatening support for their developing subject knowledge. By the act of reading the book, they were able to identify water saving strategies and practices that they reflected on in relation to their own water use, and how they would approach teaching this with YP. In addition to, this, while the trainees self-identified as not being experts in teaching about drought and water scarcity, and had concerns that teaching such a topic would raise anxiety with YP, using the book reduced these concerns and provided greater confidence and efficacy.

These findings support comments by Roche [57] that picture books can be used to support an interactive dialogue. She refers to these books as a multi-modal resource that allows for authentic dialogue. This means that using a book, such as the DRY picture book, demands real engagement through the telling of the story, thinking about the story and posing questions and answers. As such, it provides a safe space for sharing and reflection on what could be an upsetting or worrying topic (cf. concerns around climate education). Comments by teachers regarding the appropriateness of the book being for YP between 8 and 13 years old resonates with Evans [58], who argues that though picture books are illustration heavy and word light, they are not to be considered only for the very young. The illustrations in the DRY picture book confront a potential emotive response that have challenged the reader to develop a deeper understanding. This applies to the YP and the facilitator of the learning as both journey through the development of their own learning.

The analysis has shown that the DRY picture book is polysemic [58] - able to engage the reader with fundamental life issues around notions of responsibility and hydrocitizenship. This resonates with Hicks' [49] work regarding how educators can frame effective engagement with global crises and facilitate behaviour change. He notes that facilitators need to provide opportunities for learners to: acquire appropriate knowledge of the issues; explore YP's feelings towards these issues; identify relevant choices for positive change; and enable opportunities to engage in appropriate action for change.

The first three of these stages were successfully undertaken by using the book. More research is required to track further the possible impact and shift from intention towards hydrocitizenship to action for hydrocitizenship. Of interest is the way in which YP negotiated their feelings and emotions about the issues through the narrative journey. Young people demonstrated their capacity for resilience as learners. While starting with anxiety and negative feelings, these metamorphosed into the articulation of positive emotions and hope for a future that they could imagine through identified actions that promote agency. This again highlights the value of bringing water risk (deficit) alongside positive resource actions and systemic thinking. In promoting agency, the book is able to position YP as active problem solvers rather than passive recipients of information. However, arguably there needs to be change at the level of YP's positioning in society, as currently this can be

restrictive to their agency and potential. There are moral and ethical factors to be mindful of surrounding YP's agency and autonomy. Societal and family structures may not encourage YP to take positive actions in relation to hydrocitizenship leading to dashed hopes and disappointment for the child. Messages about hope and action need to be considered sensitively in the design of learning resources that cover challenging topics.

In terms of narrative approaches, Hicks [59] and Whitehouse [60] also remind us that teachers should not dwell on negative stories as this can support and develop a sense of despair, and powerlessness for our pupils in the face of global political landscapes. Instead, we need to embrace active pedagogies that will support YP to feel empowered to act as global citizens, and begin to make steps to create the future they want [22] and pedagogies promoting possibility of behaviour change as a geography of hope). The use of the DRY picture book gave space for teachers to engage in 'authentic dialogue' [58] with young learners, and so promote a real engagement and attentiveness to themes of hydrocitizenship. The YP articulated how they made links between self and the story. They were comfortable in doing so due to a culture and atmosphere of respect, trust and openness, similar to the way in which TT made sense of the picture book through reflective commentary. There is strong potential to promote hydrocitizenship and learning for resilience (water as resource and risk) in the same frame.

*6.3. Practical Applications: Where You Might Go to Next*

Alongside the DRY picture book, the development team created 'teachers' notes'; a compendium of suggested uses and activities aligned to the national curricula (e.g. [32]). This might come in the form of poster making to encourage careful water use around the school and local community based on lessons learned from the book. (DRY ran a competition themed around the DRY picture book in May 2021, linked to an annual Water Campaign). These teachers' notes provide the opportunity for the subject of drought to be taught through different curriculum subjects. For example, an English focused lesson could include developing acrostic poetry with the book as a stimulus. Poetry writing can encourage YP to use their learning and communicate messages and understandings about hydrocitizenship in new ways. Table 2 provides examples of various forms of poetry that YP in primary schools will be familiar with (haikus, acrostic and limericks). They can be a guide for adaptation for use by others—for example, water professionals or NGOs engaging YP holistically about water as risk and resource. We now plan to evaluate systematically the use of teachers' notes by teachers and other facilitators of learning. We see these as a 'scaffolding' resource that will allow more targeted and effective uses of the DRY picture book, and an opportunity to learn about drought whilst also fulfilling other curriculum requirements (though an English or Mathematics lesson). Informal feedback (at the 'About Drought Download' event, and from a national heritage NGO) suggests that such notes may also assist those outside teaching professions who may not know how to apply the book to a learning situation.

**Table 2.** Examples of acrostic poetry (nicknamed 'DRYkus') generated during the engagements with the DRY picture book.

| |
|---|
| Drought so hidden and unspoken<br>Read, learn, act, take responsibility<br>Yo—learn about precious water! |
| Drizzle, mizzle, grey days<br>Really topping up soil moisture?<br>Yellowing grass suggests not . . . . |
| Diary of a water super hero,<br>Reaching far and wide, making hopeful futures,<br>Yes! |

**Table 2.** *Cont.*

| |
|---|
| Desiccation cracks<br>Ricochet through the landscape.<br>You don't believe me? |
| Dewy drops<br>Rattling raindrops<br>Yet to arrive |
| Regret is not enough, we demand more<br>All the water is not enough to sate our thirst<br>In crisis, we'll realise<br>Not soon enough. |
| Diary of a water super hero helped me to see the<br>Reality of how we should be<br>You can help change the world by helping me |

*6.4. The Importance of Evaluation*

Evaluation forms an important part in the development of learning resources for YP. Here, we emphasise the value of multi-stakeholder evaluation, and the importance of triangulation of their perceptions. Evaluative criteria included content, impact and usability. Any intervention with YP needs to pay strong attention to their cognitive level but picture books have valuable potential to extend that range in both directions.

In exchange of research-informed outcomes targeted at YP, there are different models. We could have subcontracted the work out to a children's author or an artist. Instead here, we experienced the value of longitudinal, interdisciplinary working for knowledge exchange. Our emphasis was on co-production with interdisciplinary academics (risk geography; child psychology) working with primary educators and a socially-engaged artist in order to promote high-quality learning. Limitations of the research design include the lack of longitudinal engagements with all stakeholders. Future directions include further follow-up research with teachers and YP, and evaluation of the teachers' notes.

**7. Conclusions**

The DRY picture book as a learning resource forms a unique part of knowledge exchange about UK drought and water as a valuable resource, positioning YP as young citizens with agency. The team have organised and facilitated seminars and workshops for teachers and TT on how to use the book and its associated teaching resources. Literature and experience show that some teachers may be reluctant to engage with subjects such as drought and water scarcity as increasingly they report being concerned about rising levels of eco-anxiety and reluctance to upset YP—as well as the topics not being on (m)any educational curricula. Our observations suggest the DRY picture book is a welcome resource to enable a confident and informed approach to engaging YP with drought and the importance of valuing water. This tool provides scaffolding for teachers to support understanding of the complex issues surrounding the often hidden risk of drought in the UK. Through the narrative, visual clues and activities, the resource explores how YP can be key hydrocitizens and influencers at home and in the community. The research indicated several learning points: the value of interweaving fiction with fact in engaging YP around complex issues; the potential of creative visualisation in articulating issues around a hidden risk; and the opportunity of holistic engagements with water as both a risk and resource. The latter combines potential anxieties about a problem with potential adaptive actions to reduce its impacts. The research recognises that an (informed/trained/confident) adult may need to facilitate the understanding.

One-off engagement with a book can only go so far; however, a book can be embedded within various learning activities to encourage development of a range of competencies. In a country perceived as wet, such engagements, with young citizens and future teachers, is an important element of capacity building about increasing UK drought risk, and the

need for increased water efficiency in the context of climate change and population growth. Drought risk (and other climate related content) is not on the English curriculum and should be. There may also be a need to clarify whether the inclusion of SDGs in other UK curriculums (Wales and Scotland) leads to the inclusion of topics such as drought or whether this step (although in the right direction) is currently too broad and needs more careful definition in its integration.

**Author Contributions:** Conceptualization, L.M., V.J. and S.W. (Sarah Whitehouse); Methodology, V.J., S.W. (Sarah Whitehouse), L.M. and S.W. (Sara Williams); Formal Analysis, V.J., S.W. (Sarah Whitehouse) and L.M.; Data Curation, V.J. and S.W. (Sarah Whitehouse); Writing—Original Draft Preparation, L.M., V.J., S.W. (Sarah Whitehouse) and S.W. (Sara Williams); Writing—Review and Editing, L.M., V.J., S.W. (Sarah Whitehouse), S.W. (Sara Williams) and L.G.B.; Visualisation, V.J. and L.G.B.; Project Administration, L.M. and V.J.; Funding Acquisition, L.M. All authors have read and agreed to the published version of the manuscript.

**Funding:** UK Natural Environment Research Council Grant (No: NE/L01033X/1).

**Institutional Review Board Statement:** University of the West of England, Bristol Ethics Approval number ACE.20.06.037 Jones.

**Informed Consent Statement:** Informed consent was obtained from all subjects involved in the study.

**Data Availability Statement:** All resources are free and available online at: https://dryutility.info/learning (accessed on 3 September 2021). The book is available in English and Welsh. The development of accompanying teachers' notes aimed to support the book's effective use by educators in formal and informal learning. These are available in English, and have been edited/translated into Welsh by Natural Resources Wales. Hardcopy of the book (limited run) for educational use can be obtained by emailing: dry@uwe.ac.uk. Educators are encouraged to complete a short online questionnaire about their experience of using the resources (also available at dryutility.info/learning) and their impacts on supporting curriculum design and teacher subject knowledge. We ask for assistance in sharing the book resource around different stakeholder networks, and very much welcome feedback.

**Acknowledgments:** The contributions of the wider DRY research consortium to the DRY project are acknowledged, as are those of participants within DRY's seven catchment-based Local Advisory Groups and DRY's national Stakeholder Competency Group.

**Conflicts of Interest:** The authors have no competing interest to declare.

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
