# Peer review of "Promoting Water Efficiency and Hydrocitizenship in Young People’s Learning about Drought Risk in a Temperate Maritime Country"

_water, doi:10.3390/w13182599_

Round 1
Reviewer 1 Report
The manuscript is about one book to be used to young people (YP) and about the topic of drought risk in UK. The topic is in general relevant, but hard to be applied and explained in UK.
My main concern is about the way that the work was presented. It seems quite confused.
The manuscript should be more clear in the three stages:
1) Creating the book .Contributions and goals of the book
2) Topics in the book. Chapters subject and is the story.
3) Assessment of the book by the multi- stakeholders
On this last point, the opinions should more organised and a more statistical analysis should be presented in graphics or tables.
To the reader all these three stages should be presented in a sequential order and the conclusions should focus in more quantitative way about the performance and interest of the book for the different stakeholders.
Minor comments:
1- line388 there's a mention of three groups. Please elaborate on those and explain how they were created.
2- It is not possible to ready what is in figure 1. Enlarge it or improve it ina new format.
3- line 409, should be Figure 2.
4- In table 1 is not clear from who are the emotional responses. Why there is a jump from participant 1 to 9?
5- line 609. Please complete the caption of figure 4.
Reviewer 2 Report
First, I wish to congratulate the authors for having written a timely, interesting, and in many ways beautiful paper. It has an important topic, and the results are rich, well presented, and truly bring new information to the field. I particularly like that all three perspectives of children, trainee teachers and teachers get a voice.
My only minor comment is on this: why call 9-10-year-olds YP:s (or young people) when you could just call them children? But this is a question of style, and I suppose there is a reason for this (to my taste clumsy and unnecessary) abbreviation.
Otherwise, I cannot really think of anything to be changed for the better. Congratulations, this paper is ready to be published!
Reviewer 3 Report
Paper evaluates a pictures book that explores UK drought risk and young people’s positive water behaviours. The book’s development was underpinned by research within a project tyounth hat involves an interdisciplinary team.
A mixed method approach was used to gather perceptions about the book’s use and its impact across teachers, training teachers and young people (7-11 years old).
Some considerations that can contribute to improving the manuscript and contribute ideas of interest for a possible replication of the project and the materials in other contexts:
Clarify in more detail if the educational intervention project has had any complementary material to the book and if the intervention is associated with some type of active methodology and exhaustive curricular planning that connects the topic with curricular contents of subjects such as mathematics, language, social, artistic education...; and if there is evidence of what educational competencies are achieved with this intervention.
It would be interesting to clarify if there has been any teacher training plan with this material.
The age range from 7 to 12 is very wide, is there any adaptation in the content and language that is addressed?
There is a lack of a more explicit list of criteria (content, impact and usability) that makes it possible to evaluate the book, the methodology, the complementary materials and the program in general with the corresponding adaptations for the different ages involved.
A more general program evaluation framework would provide the manuscript with greater methodological interest.
The evaluation resulted in the need for some complementary development of resources with electronic support?
